Development, factor structure and application of the Dog Obesity Risk and Appetite (DORA) questionnaire

Raffan Eleanor 1 er311@cam.ac.uk
Smith Stephen P. 2
O’Rahilly Stephen 1
Wardle Jane 3
1 Wellcome Trust-MRC Institute of Metabolic Science, University of Cambridge , Cambridge , United Kingdom
2 Department of Clinical Medicine, University of Cambridge , Cambridge , United Kingdom
3 Health Behaviour Research Centre, Department of Epidemiology and Public Health, University College London, University of London , London , United Kingdom
Nicol Christine
Electronic publication date: 2015 Sep 29
Publication date: 2015
Volume: 3
Electronic Location ID: e1278
Received 2015 Jul 23; Accepted 2015 Sep 7
Copyright: © 2015 Raffan et al.
Copyright year: 2015
Copyright holder: Raffan et al.
License: This is an open access article distributed under the terms of the Creative Commons Attribution License, which permits unrestricted use, distribution, reproduction and adaptation in any medium and for any purpose provided that it is properly attributed. For attribution, the original author(s), title, publication source (PeerJ) and either DOI or URL of the article must be cited.
License URL: https://creativecommons.org/licenses/by/4.0/

Keywords: Eating behaviour, Questionnaire, Dog, Obesity, Owner, Factor structure, Appetite, Exercise, Feeding, Health

Funding: Medical Research Council MRC_MC_UU_12012/1 Wellcome Trust 078986/Z/06/Z This study was supported by the Medical Research Council (MRC) Metabolic Disease Unit (MRC_MC_UU_12012/1), and the Wellcome Trust grant 078986/Z/06/Z (to SOR). The funders had no role in study design, data collection and analysis, decision to publish, or preparation of the manuscript.

==============================
Background. Dogs are compelling models in which to study obesity since the condition shares many characteristics between humans and dogs. Differences in eating behaviour are recognised to contribute to obesity susceptibility in other species but this has not been systematically studied in dogs.

Aim. To develop and validate an owner-reported measure of canine eating behaviour and owner or dog related factors which can alter the development of obesity. Further, to then test variation in food-motivation in dogs and its association with obesity and owner management.

Methods. Owner interviews, a literature review and existing human appetite scales were used to identify relevant topics and generate items for the questionnaire. Following a pilot phase, a 75 item online questionnaire was distributed via social media. Responses from 302 dog/owner dyads were analysed and factor structure and descriptive statistics calculated. Results were compared with descriptions of dog behaviour and management from a subset of respondents during semi-structured interviews. The optimum questions were disseminated as a 34 item final questionnaire completed by 213 owners, with a subset of respondents repeating the questionnaire 3 weeks later to assess test–retest reliability.

Results. Analysis of responses to the final questionnaire relating to 213 dog/owner dyads showed a coherent factor structure and good test–retest reliability. There were three dog factors (food responsiveness and satiety, lack of selectivity, Interest in food), four owner factors (owner motivation to control dog weight, owner intervention to control dog weight, restriction of human food, exercise taken) and two dog health factors (signs of gastrointestinal disease, current poor health). Eating behaviour differed between individuals and between breed groups. High scores on dog factors (high food-motivation) and low scores on owner factors (less rigorous control of diet/exercise) were associated with obesity. Owners of more highly food-motivated dogs exerted more control over their dogs’ food intake than those of less food-motivated dogs.

Conclusions. The DORA questionnaire is a reliable and informative owner-reported measure of canine eating behaviour and health and management factors which can be associated with obesity development. The tool will be applicable to study of the canine obesity model and to clinical veterinarians. Results revealed eating behaviour to be similarly associated with obesity as exercise and owners giving titbits.

Introduction

Obesity is a common problem in dogs and is associated with a plethora of diseases and physiological changes that are reminiscent of those seen in humans (Edney & Smith, 1986; German, 2006; Zoran, 2010). Dogs share an environment with their human owners and are exposed to similar obesogenic factors such as a sedentary lifestyle and ready availability of calorie-dense, highly palatable food (Raffan, 2013). Consequently, canine obesity is worthy of study not only to address veterinary questions but also because dogs are compelling model organisms for research of translational relevance (Raffan, 2013).

Obesity is ultimately the consequence of energy intake exceeding expenditure. However, there is an increasing recognition that energy homeostasis is a closely regulated neurobiological process under the influence of genetic variants which often modify an individual’s behavioural phenotype related to feeding style, energy intake and food seeking behaviour (Carnell & Wardle, 2009; Farooqi, 2011; Raffan, 2013; Yeo & Heisler, 2012). People demonstrate marked variability in appetite and food preferences. Many studies have shown that variation in eating styles is associated with obesity. Genetic, endocrine and psychological mechanisms for this association have been explored (French et al., 2012).

There is evidence from experimental feeding trials that variability in eating behaviour exists between individual dogs and breeds (Hewson-Hughes et al., 2013) and a single veterinary study reported appetite (tested with a single question) to contribute to dog obesity (Sallander et al., 2010). However, individual feeding experiments are expensive and time consuming, as are similar protocols applied to human research subjects (Gibbons et al., 2014).

Consequently, questionnaire based tools to assess eating traits such as food responsiveness and satiety responsiveness in people are well established (Llewellyn et al., 2011; Van Strien et al., 2006; Wardle et al., 2001). No such tool has been reported for dogs and we identified the need for one to assess differences in eating behaviours between individual dogs. The role of dog owners as caregivers with a close emotional bond to their wards is reminiscent of that of parents with children so we considered questionnaires about eating behaviour for children to be a good model for developing such a tool.

Questionnaires for children and babies drew upon established theories and research (Llewellyn et al., 2011; Wardle et al., 2001). No such systematic research examining dog eating styles exists on which this questionnaire might be based. However, we can infer that variation exists because animal care texts refer indirectly to eating styles by advising on how to manage dogs with food related behaviours which are considered desirable or not. Additionally, food is commonly used as positive reinforcement during training (Hand et al., 2010; Lindell, 2009).

Since dog owners are generally responsible for providing food and enabling exercise for their pets, it is sensible that a questionnaire considering the influence of food-motivation on obesity should also collect information about owner or dog factors which might moderate the development of obesity. Obesity in dogs has previously been associated with gender, neuter status, age, breed, feeding practices (e.g., provision of human food, titbits, weighing/measuring food), owner income, obesity and age, the nature of the owner-dog relationship and exercise (Bland et al., 2009; Colliard et al., 2006; Courcier et al., 2010; Edney & Smith, 1986; Kienzle, Bergler & Mandernach, 1998; Nijland, Stam & Seidell, 2010; O’Neill et al., 2014; Robertson, 2003; Rohlf et al., 2010; Sallander et al., 2010; Warren et al., 2011).

Only one previous study (Sallander et al., 2010) considered appetite or food motivation, and then only with a single question. However, anecdotally, veterinary surgeons and pet owners commonly report some individual dogs or breeds/types of dog to be particularly food motivated or ‘fussy’ eaters.

The primary aim of this work was to develop an owner-reported questionnaire to assess dog eating behaviour as well as dog health and owner management factors which can affect the development of obesity and validate it using different measures (test test–retest reliability, internal consistency, face validity, construct validity and criterion validity).

A secondary aim was to use the questionnaire to test whether variability in dog food-motivation existed and, if so, how it was associated with different owner management styles, and obesity. The null hypotheses were that there would be no difference in eating behaviour between dogs or between lean and overweight dogs, and that owners would not manage food differently in dogs with different eating behaviours.

Materials and Methods

The study was approved by the Ethics and Welfare Committee of the Department of Veterinary Medicine, University of Cambridge (CR125).

Questionnaire development

Defining variability in food related behaviour and owner management

Items to be included in the questionnaire were designed using information from three sources:

(1) A minority of items were drawn directly from two existing questionnaires for assessing eating behaviour in children and babies (Llewellyn et al., 2011; Wardle et al., 2001). Existing items were selected if they could be deemed relevant to dogs. For instance, some items could be adapted by substituting ‘dog’ for ‘child’, such as ‘my child finishes his/her meal very quickly’. Others were rejected as they asked about interactions not recognised or defined in dogs, such as ‘I use puddings as a bribe to get my child to eat his/her main course’. Furthermore, these resources were used to identify a list of topics which might be relevant to dogs or owner management. Amongst those topics was ‘emotional eating’ which we defined as eating more or less food during negative emotional states, following Wardle et al. (2001).

(2) Veterinary texts and clinical experience were used to identify further areas where feeding behaviours were variable between dogs, and where owners were likely to make decisions about how they interact with their dogs with regard to food. Dog owners have commonly expressed concern to one author (ER, a veterinary surgeon) about behaviours they considered undesirable such as very fast eating, fussiness or choosiness, ‘begging’ for food, and snatching food, or discussed other behaviours as desirable such as waiting patiently to be fed, eating only on command, and taking proffered titbits gently. Other behaviours such as food-related aggression, stealing human or other pets’ food, coprophagia, or eating non-food items such as stones or socks are also more formally addressed (Lindell, 2009). Additionally, food is advised as a training aid (Lindell, 2009). Literature reviews were also used to identify known factors which predispose dogs to obesity (see ‘Introduction’).

(3) Findings from (1) and (2) were used to generate a framework for semi-structured interviews. Interviews with an opportunistic sample of dog owners who had volunteered their dogs for participation in a different genetic research project were performed by a single author (ER) who is an experienced veterinary clinician. Detailed, hand-written contemporaneous notes were made during and immediately after each interview.

Initially, owners were encouraged using open questions to talk about the eating behaviour and feeding, exercise and other management of a single dog they owned. Comparisons between other dogs they knew or owned were encouraged, to draw out differences in eating behaviour. Interviews were conducted non-judgementally and presented as being about natural variation in dog behaviour, to encourage owners to be as candid as possible about their management practices. The interviewer avoided pejorative terms such as ‘greedy’ and ‘begging’. Areas addressed specifically in the semi-structured interviews were broadly divided into ‘dog’ and ‘owner management’ topics which are summarised below. A copy of the primer used for the interviews is included as File S1.

Dog topics were: (a) responsiveness to food (behaviour at mealtimes or when food offered in between meals, tendency to inspect food before eating), (b) speed of eating (of meals or offered titbits), (c) satiety responsiveness (interest in food in between meals, apparent desire for more food than usually offered), (d) general interest in eating (food seeking behaviour on walks, behaviour when human food was being prepared or eaten), (e) food fussiness (selectivity about particular foods, refusal to eat some types of dog food, choosiness about which titbits to eat), (f) emotional eating (change in appetite in response to owner-perceived changes in a dog’s stress levels), (g) scavenging/stealing food (eating items owners consider revolting such as faeces, indigestible such as stones, or illicit such as human food left within reach but in places they are taught not to access), and (h) food related aggression (to humans or other dogs).

Owner management topics were: (a) control over eating (regulation of food intake by offering measured amounts at mealtimes, regulating titbits, and stopping scavenging or stealing), (b) prompting/encouragement (offering multiple diets or addition of flavoursome foods, or encouragement given to dogs to eat), (c) instrumental feeding (use of food rewards in training), (d) emotional feeding (whether they fed their dogs to provide reassurance), (e) importance of body shape (how important owners felt it was to manage their dogs’ weight, and how hard they perceived they worked to do so), (f) exercise (importance attributed by owners to exercise, nature, duration and frequency of exercise in a typical week), and (g) level of training (whether dogs were trained for a specific purpose, level of discipline, ability to stop undesirable behaviours successfully, desire to control begging/scavenging or other undesirable behaviour). Owners were asked specifically about their dogs’ health (including questions about dietary sensitivities, or continuous or intermittent clinical signs of gastrointestinal upset) in order to identify any actual or suspected disease which could affect appetite or eating behaviour.

Generation of items for questionnaire

Systematic analysis of interview data was performed by manual coding of copies of the contemporaneous notes from each interview by a single investigator who also performed the interviews (ER) (Gibbs, 2007). Coding of the information was categorical and both concept driven (some codes generated from literature review as above) and data driven (additional codes which emerged from interview data) (Gibbs, 2007). Following coding, the frequency with which each category was coded in the interview data was used to prioritise how many associated items should be included in the pilot questionnaire.

The repeated review of the data was also used to extract recurring phrases and descriptors used by owners to describe their dogs or management practices. Final items were constructed using that information, coding data and adaptation from existing human questionnaires.

Questionnaire refinement and sampling

Below, the different versions of the questionnaire used during the process of development are outlined, along with a brief summary of what was tested at each stage. Later sections address the sampling methods and statistical methods applied at each stage. A flow diagram to summarise this process is shown in Fig. 1.

Figure 1 Flow diagram summarising validation and application steps for the DORA questionnaire.

Pilot survey—identifying initial problems and testing face validity

Seventy-seven items were included in a preliminary questionnaire with a single response format (‘never’, ‘rarely’, ‘sometimes’, ‘often’, and ‘always’). Items related to dog eating behaviour and owner management were mixed in random order. ER discussed each item with a panel of experts including co-authors (JW and SS) and 4 veterinary colleagues with an interest in behaviour. Dog owners (sample 1) were invited to test the pilot questionnaire either online or on paper. Those who completed a paper copy and were encouraged to discuss the questions (particularly any areas where they were unclear what was being asked or weren’t sure how to answer for their dog) with an investigator as they did so and those who completed it online were invited to make a text comment after each item and talked to an investigator afterwards. Contemporaneous notes were made on all discussions. Responses were critically evaluated to test face validity (i.e., did different respondents interpret items in the same way as each other and/or in the way intended when they were composed?).

Factor structure and questionnaire refinement

A refined 75 item questionnaire was published and disseminated online (sample 2). In addition to the items composed above, the name, age, breed and gender of each dog was recorded. Owners were invited to assign their dog a body condition score (a numerical, categorical scale of dog body morphometry which has been shown to be a reliable indicator of body adiposity and takes into account the variability of body shape in different breeds) on a scale of 0–5 where 0 is excessively thin, 3 optimal and 5 obese, guided by a chart with representative images and accompanying descriptors (Laflamme, 1997; McGreevy et al., 2005). Questions about breed, gender and body condition score were not compulsory. Respondents also had the option of providing their name and contact details on the understanding that some would be contacted to be interviewed about the subject of the survey.

Factor analysis was performed and responses to individual questions statistically described (see statistics section). Of respondents who had provided contact details, a subset were contacted by telephone and interviewed using the same semi-structured format as previously, but with additional questions about particular answers which appeared discordant with other questionnaire responses.

The number of questions was reduced to minimise the length and avoid clearly repetitive items. Questions which had a broad range of answers and loaded onto just one factor were prioritised (see statistics section).

Testing the refined questionnaire

The resulting shorter questionnaire was distributed online (sample 3). Three weeks after they had taken the questionnaire, respondents were invited to repeat it. Factor analysis and analysis of test–retest reliability were performed (see statistics section).

Method of dissemination

At each stage, the online version of the questionnaire was composed, formatted and published using commercially available software (Qualtrics, Provo, USA). The survey software prevented more than one submission per internet provider (IP) address, and respondents with more than one dog were asked to answer about only one dog. For online dissemination, posts were made on Facebook and Twitter, and interested parties including leaders of rescue organisations, the Kennel Club, dog-related businesses and friends and colleagues of the authors were encouraged to disseminate the invitation to participate more widely.

Sampling

Sample 1 (pilot questionnaire) was a small convenience sample of dog owners who were colleagues or associates of ER.

For sample 2 and 3, the questionnaire was disseminated exclusively online and thus they can be considered convenience samples drawn from a large pool of dog owners active on social media.

A subset of sample 2 were contacted by telephone to discuss their answers to the survey. Those individuals were initially self-selecting (because they opted to give their contact details). From those owners, owners were chosen to be representative of a range of differently scoring dogs of different breeds and ages, from different parts of the country.

All owners in sample 3 who had given their contact details were invited to retake the questionnaire 3 weeks later and all responses received were analysed. Responses were not permitted from respondents who had taken previous versions of the questionnaire.

Questionnaire scoring system

Since both 4 and 5 option scales were used, items were scored as a percentage of the maximum to avoid 5 option scaled questions having a greater impact than 4 option scaled questions. Therefore, ‘never’ was assigned 0, ‘rarely’ 25, ‘sometimes’ 50, ‘often’ 75, and ‘always’ 100. Similarly, ‘not at all true’ was assigned 0, ‘somewhat true’ 33.3, ‘mainly true’ 66.6, and ‘definitely true’ 100. Where items within a factor invited opposite answers for the same dog, scores for the minority items were reversed. (For instance, within a theoretical factor about dog size, ‘true’ for ‘my dog is very small’ would score 100 as would ‘false’ for ‘my dog is very big’.)

The working scores for the questionnaire were the individual factor scores, a combined ‘dog food-motivation score’, and a combined ‘owner management score’.

Individual factor scores were calculated as percentages:

= (sum of item scores for that factor)/(sum of maximum possible item scores for that factor) × 100.

The dog food-motivation score was calculated from all the dog items:

= (sum of item scores for all dog items)/(sum of maximum possible dog item scores) × 100.

The owner management score was calculated from all items in factors 1–3 of management and health factors (i.e., those related to food control and perception of dogs’ weight):

= (sum of item scores for all items in factors 1–3)/(sum of maximum possible item scores in factors 1–3) × 100.

Statistical analysis

All numerical analyses were conducted using the R statistical programming language (R Core Team, 2013).

Summary of validation tests

Different validation methods as applied to veterinary questionnaires have recently been reviewed and definitions were applied as in that paper (Belshaw et al., 2015).

Test–retest reliability was measured on the final version of the questionnaire.

Face validity (a type of content validity) was assessed during the pilot survey stage by interrogating an expert panel and dog owners on their interpretation of items forming part of the questionnaire.

Results of factor analysis, the Tucker Lewis index (a measure of factoring reliability) and Cronbach’s alpha are reported as measures of internal consistency.

Construct validity was assessed by testing for hypothesised relationships between different factors: we hypothesised that the three dog factors would be positively correlated with one another and with the overall dog food-motivation score; also that owner factors related to food control would be correlated with one another and with the overall owner management score.

Criterion validity for known groups was assessed by testing whether dog food-motivation scores were different between different dog breeds, and dogs with different body condition scores. Furthermore, we hypothesised that factors which encompassed previously reported risk factors for obesity would be positively associated with body condition score.

Exploratory factor analysis (samples 2 and 3)

Parallel and factor analysis algorithms were used, as implemented in the Psych R package (Revelle, 2015). Results from parallel analysis informed the number of factors analysed subsequently in exploratory factor analysis. Factor loadings of greater magnitude than 0.4 were considered substantial. Plotting functions were implemented in the ggplot2 package (Wickham, 2009). The Tucker-Lewis index of factoring reliability and Cronbach’s alpha were calculated using the functions in the Psych R package. For the Tucker-Lewis index, 0.9 was considered to indicate good factoring reliability. For Cronbach’s alpha, results >0.6 were considered acceptable, >0.7 good and >0.8 very good.

Reducing items—refining the questionnaire

After factor analysis of Sample 2, questions which had a broad distribution of answers (low skew), loaded onto just one factor (low complexity), and much of whose variability could be explained by the factors identified (high communality) were prioritised. All three were calculated in the Psych R package as part of exploratory factor analysis (see above).

Skew is a measure of the asymmetry of the probability distribution of a real-valued random variable about its mean. Low skew was considered desirable and items with skew >1.25 or <−1.25 were considered poor. Complexity refers to the number of factors upon which a variable has moderate or high loadings. Complexity <2 was considered desirable and items with complexity >3 were considered poor. Communality refers to the proportion of variance in each variable which can be explained by the factors. Communality greater than 0.5 was considered desirable and items with communality <0.35 were considered poor.

Validating the final questionnaire and testing the secondary aims

Final validation and testing the hypotheses defined as secondary aims were performed on the data from Sample 3. i.e., the results from the final version of the questionnaire. Descriptive statistics were generated for all factors. Data for all factors and other variables were tested for normality of distribution by measuring skew and kurtosis, and by graphical analysis using frequency distribution and quantile–quantile plots. Methods for specific tests applied at this stage are detailed below.

Test–retest reliability

Test–retest reliability was analysed using the cor.test function in R, which calculates test statistics to measure the significance of correlation between scores for each factor at the different time points. For correlation coefficients, R, >0.5 was considered good and >0.75 very good. For interpretation of P values, see below.

Correlation

When associations between quantitative variables (e.g., factor scores, body condition scores, age) are reported, correlation was tested using Pearson correlation for normally distributed data and Spearman correlation for data which were not normally distributed. Correlation coefficient, R, > 0.5 was considered good and >0.75 very good.

Tests to compare groups

Intergroup comparisons were assessed using ANOVA (e.g., for breeds) or paired t-tests (e.g., for gender).

Generation of a minimum model to explain body condition score

Stepwise multiple regression to a minimum model was performed in R using factor scores as predictors of body condition score with a final model defined when all remaining factors were significant independent predictors.

Significance levels

For comparisons involving age, gender and breed, testing the hypotheses defined in the introduction, assessing test–retest reliability and defining the minimum model during stepwise multiple regression, significance was determined by the test statistic p < 0.05. Since testing for correlations between dog and management/health factors involved multiple testing, a Bonferroni corrected level of significance of p < 0.001 was used for all those comparisons.

Results

Questionnaire development

Defining variability in food related behaviour and owner management

During preliminary interviews owners were keen to talk about their dogs and commonly volunteered information about many or all of the topics in the interview framework. Many owners described dog eating behaviours in pejorative and frank ways. For example, dogs were commonly described as ‘greedy’. In contrast, they tended to avoid pejorative terms for behaviours which textbooks commonly describe as suitable for modification by training. For example, owners would commonly describe behaviours such as hanging around at human mealtimes, or using eye gaze direction to identify food. Owners interpreted those behaviours as soliciting food but would deny that their dog ‘begged’ for food if asked directly. These findings were taken into account when designing items for the questionnaire.

Generation of items for questionnaire

Most of the topics related to eating behaviour and owner management emerged during owner interviews and recurrent phrases were used to write items for the questionnaire. Of 34 codes applied to the data, some recurred frequently, such as differences in selectivity (for example, ‘my dog will eat anything’ or ‘my dog is interested in human food but only actually eats the things he likes’). Other codes were not commonly applicable and hence not represented in the questionnaire. For instance, only 1 out of 50 dog owners reported a difference in their dog’s eating behaviour during periods of stress (interpreted as ‘emotional eating’) and none reported feeding their dogs to provide comfort when upset, although use of food as a reward for good behaviour after a stress such as a veterinary visit was common, which might be viewed similarly. Following analysis, 77 items were written for inclusion in the pilot questionnaire.

Sampling

For sample 1 (pilot survey), completed questionnaires were received from 22 dog/owner dyads representing 10 breeds. Fifteen completed a paper copy and 7 completed the questionnaire online. Mean (SD) age was 6 years (3.8).

For Sample 2, the questionnaire was started by 298 owners but 78 (26%) failed to finish within 2 weeks meaning completed questionnaires were analysed from 224 dog/owner dyads. Dogs had mean (SD) age of 6 years (3.5). Labradors predominated in this sample (n = 159, 86%). Seventeen crossbreed dogs and 7 or fewer dogs from 23 other breeds completed the group.

Email addresses were provided for 204/224 owners in Sample 2 who completed the survey and a subset of 20 were contacted by telephone for follow-up interview.

For Sample 3, the questionnaire was started by 244 owners but 39 (16%) failed to finish within 2 weeks meaning completed questionnaires were analysed from 205 dog/owner dyads. The median time to complete the entire questionnaire was 9 min (mean 15 min). The mean (SD) dog age was 6 years (3.8) and median age (25th and 75th centiles) 6 years (3, 9). Thirty-seven breeds were represented, including 40 cross-breeds, 101 Labrador retrievers, and 6 or fewer dogs from each of 34 other breeds. There were 106 male dogs (4 entire, 10 neutered, 92 not reported), 79 female dogs (2 entire, 17 neutered, 60 not reported) and 20 with gender not reported.

Email addresses were provided for 185/205 owners in sample 3 who completed the survey and all were invited to take the survey again 3 weeks later. Sixty-seven complete responses were received within 7 days and were analysed for test–retest reliability.

For both Samples 2 and 3, Labrador retrievers were the predominant breed. This was found to be related to prominent posts on social media by the head of a Labrador retriever rescue organisation. Labrador retrievers are estimated to make up 10–16% of the UK dog population (Asher et al., 2011) meaning the breed was significantly (Chi squared test, p < 0.001) over-represented in both samples.

Pilot survey—identifying initial problems

After the pilot stage (Sample 1) assessment of face validity, two items were discarded, and wording was modified to increase clarity for others. The response format was changed for 48 of the remaining items to ‘not at all true’, ‘somewhat true’, ‘mainly true’, and ‘definitely true’ because testers felt that format better suited particular items.

Factor structure

Data from Sample 2 were tested using exploratory factor analysis. Of the 75 items, all but 6 were associated with one or more of 9 factors. Three factors were related to dog eating behaviour, 4 to owner management and two to dog health. Although the factors were reminiscent of the 8 dog, 7 owner and 1 health topics identified in the methods and examined during interviews, there were fewer factors and there was significant overlap; that is, topics which were a priori considered separately did not always separate into different factors using this unsupervised statistical approach.

Reducing items—refining the questionnaire

There were 23 items with skew >1.25 or <−1.25. Of these, 5 were retained due to their importance in other regards and the remaining 18 discarded. Low communality was identified for 20 items; of these 2 were retained and 18 discarded. High complexity was identified for 21 items; of these 4 were retained and 17 discarded. All 6 items which were retained despite falling outside the above categories are shown in Table 4. Overall, 41 questions were eliminated (several failed for more than one reason), leaving 34 in the final questionnaire.

Questions with high skew were typically included in the pilot survey for completeness despite not having emerged from owner interviews (e.g., ‘I think my dog could do with gaining some weight’) or had emerged during owner interviews but were rarely positively answered across a larger population (e.g., ‘My dog grazes at the food in his/her bowl throughout the day’).

Questions with low communality were typically querying management factors which did not directly impact on the factors identified (e.g., ‘My dog’s exercise varies throughout the week’ captured variability that exists in some dogs’ lives but other items better captured the amount or intensity of exercise they received).

Questions with high complexity typically used language that was hard to define or were about behaviour that varied over time (e.g., ‘My dog is lazy’).

Validating the final questionnaire

Internal consistency

For the final version of the questionnaire (Sample 3), the factor structure identified previously was maintained (Tables 1 and 2). The Tucker–Lewis index of factoring reliability was 0.94 for dog factors and 0.97 for owner management factors. The factor structures accounted for 58% of variability for the dog factors, and 62% for the owner management and health factors. Cronbach’s alpha was good or very good for all factors (Table 3).

Table 1 Factor loadings for all dog food-motivation factors in the questionnaire.

	Factor analysis	
Itema	Food responsiveness and satiety	Choosiness	Interest in food	
Dog factor 1: food responsiveness and satiety				
My dog gets excited when there is food around	0.63		0.29	
My dog will turn down food if s/he is not hungry (R)	−0.70	0.21		
My dog finishes a meal straight away	0.96		−0.17	
After a meal my dog is still interested in eating	0.52	−0.14	0.38	
My dog takes his her time to eat a meal (R)	−0.84			
My dog seems to be hungry all the time	0.41		0.33	
My dog is very greedy	0.54	−0.10	0.37	
Dog factor 2: lack of fussiness				
My dog inspects unfamiliar foods before deciding whether to eat them (R)		0.79	0.12	
My dog is choosy about which titbits he eats (R)		0.71		
My dog would eat anything		−0.79		
Dog factor 3: interest in food				
My dog hangs around for titbits even if there is not much chance of getting them			0.79	
My dog hangs around when I am preparing or eating human food	−0.12		0.75	
My dog eats titbits straight away	0.19	−0.27	0.32	
Notes.

a Items marked (R) have been reversed for scoring purposes.

b All factor loadings of greater magnitude than 0.1 are shown and those >0.4 are shown in bold.

Table 2 Factor loadings for all management and health factors in the questionnaire.

	Factor loadingsb	
Itema	Owner perception	Owner interventions	Human food	Exercise 1d	Exercise 2 d	Signs GI disease	
1f: owner perception							
I am happy with my dog’s weight (R)	0.81						
I think my dog could do with losing some weight	−0.92						
My dog is very fit (R)	0.52			0.43 c			
2f: owner intervention to control weight							
I am careful to regulate the exercise my dog gets in order to keep him her slim	−0.13	0.47 c		0.4 c			
I alter the food my dog gets in order to control his her weight	−0.28	0.6					
I am careful about my dog’s weight	0.17	0.75					
I weigh or measure how much food I give my dog		0.49					
3f: restriction of human food							
My dog gets no food at human mealtimes			−0.69				
My dog gets human leftovers in his her food bowl (R)		−0.1	0.59				
My dog gets bits of human food when we are eating (R)			0.85		0.13		
My dog often gets human food (R)			0.78		−0.13		
4: exercise taken							
My dog runs around a lot	0.16			0.72			
My dog’s walks involve a lot of energetic play or chasing				0.72	−0.15		
My dog gets a lot of exercise				0.71			
My dog’s walks are mostly on the lead (R)					0.88		
My dog spends most of his her walks off the lead					−0.9		
5: signs GI disease							
My dog gets an upset tummy on some foods	0.11					0.78	
My dog has a sensitive stomach						0.88	
My dog often gets tummy upsets		−0.1		0.11		0.73	
6: current disease							
My dog regularly sees the vet for health problemse	−0.12	0.14				0.11	
I restrict my dog’s exercise because of veterinary advicee	−0.13		0.11	−0.12	0.24		
Notes.

a Items marked (R) have been reversed for scoring purposes.

b All factor loadings of greater magnitude than 0.1 are shown and those >0.4 are shown in bold.

c These items loaded onto more than one factor, but were grouped where they loaded most strongly.

d Although factors Exercise 1 and Exercise 2 were identified separately under factor analysis, they were considered as a single factor for scoring.

e These questions did not load onto any factor, which is not surprising given that they are not directly weight, food or exercise related. However, they were retained in the questionnaire as a way to flag up the possibility that a dog has a chronic health problem which might in some way confound the weight or appetite phenotype.

f Factors 1–3 were combined to generate the ‘owner management score’.

Table 3 Test–retest reliability and internal consistency.

Data based on sample 3 respondents who completed the questionnaire on two occasions, two weeks apart (n = 67). Shown are the correlation coefficient, R, and significance, P between scores for each factor at the different time points. Cronbach’s alpha, a measure of internal consistency, is shown for each factor.

	Test–retest reliability	Internal consistency	
	R	P value	Cronbach’s alpha	
Dog factors				
Food responsiveness and satiety	0.95	>1 × 10−9	0.91	
Choosiness	0.89	>1 × 10−9	0.81	
Interest in food	0.81	>1 × 10−9	0.73	
Management factors				
Owner perception	0.57	4 × 10−7	0.86	
Owner intervention	0.80	>1 × 10−9	0.65	
Human food	0.71	>1 × 10−9	0.80	
Exercise 1	0.86	>1 × 10−9	0.77	
Exercise 2	0.81	>1 × 10−9	0.89	
Signs GI disease	0.78	>1 × 10−9	0.83	

Table 4 Questions retained despite high skew, low communality or high complexity.

Numbers in normal font are within the pre-defined boundaries for inclusion. Numbers in bold are outside those boundaries and would have been discarded as routine but items were retained for a particular reason or reasons.

	Parameter (cut-off for exclusion)		
Item retained	Skew (<−1.25 or >1.25)	Communality (<0.35)	Complexity (>3)	Reason for retention	
My dog eats titbits straight away	−1.58	0.38	2.70	Skew suggests most owners report this item was not true about their dog. During semi-structured interviews this was identified as a strong discriminator between dogs with high food-motivation and those which were notably ‘picky’ or ‘fussy’ with food.	
My dog finishes a meal straight away	−1.66	0.69	3.11	As above.	
I think my dog could do with losing some weight	1.33	1.02	0.89	Echoed phrase used by interviewees who acknowledged dog was overweight. Skew expected given that only a proportion of dogs are genuinely overweight.	
My dog often gets human food	1.22	0.29	3.3	Next best performer in this factor and considered important because did not specify when the dog was given human food (c.f. other items in factor relating to mealtimes, leftovers, etc.)	
My dog is very fit	−0.79	0.50	3.1	Echoed phrase commonly volunteered by interviewees with lean dogs and next best performer in the ‘Owner perception’ factor.	
I restrict my dog’s exercise because of veterinary advice	3.65	0.46	1.32	A warning sign of ill health or orthopaedic disease of potential value to researchers.	
My dog has a sensitive stomach	1.27	0.21	3.5	A warning sign of possible GI disease of potential value to researchers.	

Test–retest reliability

Test–retest reliability was good or very good for all factors (Table 3).

Construct validity

For dog factors, correlation between individual factors and the overall dog food-motivation score was high (Table 5A); as a consequence the composite ‘dog food-motivation score’ was used as a single measure of eating behaviour. Correlation was low-moderate for management factors when compared pair-wise but good when compared with a combined ‘owner management score’ (Table 5B).

Table 5 Spearman correlation coefficients for scores for dog and owner management factors.

(A) dog factors and (B) management factors. A Bonferroni corrected level of significance of p < 0.001 was used. Significant correlations are shown in bold and non-significant correlations are shown in parentheses.

A: dog factors	Food responsiveness and satiety	Lack of fussiness	Interest in food	Dog food-motivation score	
Food responsiveness and satiety	1	0.678	0.503	0.947	
Lack of fussiness		1	0.429	0.826	
Interest in food			1	0.667	
Dog food-motivation score				1	
B: management factors	Restriction of human food	Owner perception	Owner intervention to control weight	Combined owner management score	
Restriction of human food	1	(−0.059)	(0.139)	0.395	
Owner perception		1	(0.142)	0.520	
Owner intervention to control weight			1	0.839	
Combined owner management score				1	

The finding of only low-moderate correlation between ‘owner perception’, ‘owner intervention to control weight’ and ‘restriction of human food’ prompted us to review the semi-structured interview notes and explained how although the overall scores were associated, there was less clear association between individual factors. To explain this, we offer two example owners: Owner 1 reported they had a fit dog of good weight (scoring low in ‘owner perception’) but achieved that by very careful management (scoring high for ‘owner intervention’) but with a feeding regime that included regular human titbits (scoring moderately for ‘human food’). Overall, such owners would typically score highly on the owner management score. In contrast, Owner 2 recognised their dog was overweight (scoring high in ‘owner perception’), fed their dog ad libitum (scoring low for ‘owner intervention’) and included regular human titbits (scoring moderately for ‘human food’). Overall, such owners would typically have a low owner management score. However, on the ‘human food’ factor alone, they scored similarly to the first example owner.

Criterion validity

Factors identified which represented previously considered risk factors for obesity in dogs were ‘Exercise taken’ and ‘Restriction of human food’. The hypothesis that these would be associated with body condition score was confirmed: there were significant correlations between high body condition score and low scores for restriction of human food (Pearson r = 0.1867, p = 0.0157) and low exercise scores (Spearman r = − 0.29, p = 0.0002). There were also significantly different dog food-motivation scores for dogs of different breed groups and body condition scores (see below for further information).

Questionnaire findings

Dog characteristics

Findings are presented from Sample 3 for which the questionnaire was completed for 205 dog/owner dyads. Owners reported body condition scores for 165 dogs. Body condition scores ranged from 1 to 5 and were normally distributed with a strong peak at the optimum body shape (Fig. 2).

Figure 2 Frequency histogram showing distribution of owner assigned body condition scores.

Variability in canine appetite

Dog food-motivation scores varied widely between individual dogs, ranging from 9 to 100% with a bimodal distribution (Fig. 3).

Figure 3 Frequency histogram showing distribution of dog food motivation scores.

If Labrador retriever dogs were removed from the analysis, the distribution pattern remained almost identical.

There was a weak (r squared 0.146) but statistically significant (p = 0.037) correlation between age and dog food-motivation score. There was no difference between dog food-motivation score between male and female dogs. Neuter status was only available for a small subset of dogs, limiting the utility of analysis of the effect of neuter status, but mean dog food-motivation scores were higher in neutered dogs (mean ± SEM of 6 entire dogs 58.67 ± 9.687 vs. 77.11 ± 3.211 for 27 neutered dogs, p = 0.0304).

Given the small numbers for some breeds, British Kennel Club groupings were used for inter-breed analysis. Within the gundog group, 101/118 were Labrador retrievers. There was a significant difference between dog food-motivation scores between breed groups (Fig. 4, ANOVA p < 0.0001).

Figure 4 Box and whisker plot showing difference in dog food-motivation scores between breed groups.

Boxes extend to 25th and 75th percentiles, central line shows median, whiskers the minimum and maximum values.

Owner management, exercise and health

Owner management scores were widely and approximately normally distributed (Fig. 5A). Exercise scores were reasonably widely distributed but were slightly skewed towards 100%, suggesting the majority of dogs were considered by their owners to be very active (Fig. 5B).

Figure 5 Frequency histograms showing distribution of factor scores.

(A) owner management, (B) exercise, (C) current disease and (D) signs of GI disease scores.

The scores for current disease and signs of gastrointestinal disease were skewed strongly, showing most dogs had no persistent health problems or signs of gastrointestinal disease (Figs. 5C and 5D). There was no difference in body condition score or food-motivation score between dogs whose owners indicated they had any signs of gastrointestinal disease and those whose owners indicated they had no signs of gastrointestinal disease. Similarly, there was no significant correlation between gastrointestinal disease score and dog food-motivation score (correlation coefficient R = 0.033, p = 0.64). Dogs which scored 0 for current disease (indicating no regular veterinary visits or veterinary advised restriction of exercise) had significantly lower body condition scores than dogs which had any positive answers to those questions; there was no difference in food-motivation between the two groups. Similarly, there was no significant correlation between current disease score and dog food-motivation score (correlation coefficient R = 0.037, p = 0.59). A post hoc power calculation showed that a sample size of 200 had 80% power to detect a significant (α = 0.05) correlation greater than 0.175.

Obesity, eating behaviour, and owner management

High dog food-motivation scores were significantly associated with higher body condition scores (Fig. 6). Dog food-motivation score was positively correlated with owner management score (Fig. 7A). Consistent with the effect of high food-motivation on body condition score, there was also a positive correlation between body condition score and owner management score (Fig. 7B).

Figure 6 Scatter plot showing difference in dog food-motivation scores between dogs with different owner-assigned body condition scores.

Median and standard deviation superimposed as lines. (ANOVA p = 0.0041.)

Figure 7 Owners of more food-motivated or overweight dogs made a statistically significantly greater effort to control their dogs’ food intake.

(A) Scatter plot showing correlation between dog food-motivation and owner management scores. Spearman correlation rs = 0.30, p = 0.00001. (B) Box and whisker plot showing difference in owner management scores between dogs with different body condition scores. Boxes extend to 25th and 75th percentiles, central line shows median, whiskers the minimum and maximum values. Pearson correlation coefficient r = 0.19, p = 0.01.

As stated above (under Criterion Validity), there were significant correlations between high body condition score and low scores for restriction of human food and low exercise scores.

Minimum model to predict body condition score

Stepwise multiple regression to minimum model showed the following significant predictors of body condition score: food responsiveness & satiety (positive, p = 0.02), exercise (negative, p < 0.01) & restriction of human food (negative, p = 0.03). Effect magnitudes were similar for each predictor (0.018, 0.019, 0.016 respectively) implying a similar contribution to the body condition score for each factor.

Discussion

The primary aim of this research was to design an owner-reported measure of dog food-motivation, owner management and dog health factors which influence body weight. We report the development, validation and application of the DORA questionnaire, and show it is a robust and widely applicable tool suitable for use in research of the canine obesity model, or as a veterinary clinical tool. Below, the factor structure, methods of development and validation, and comparisons with similar human questionnaires will be discussed. Subsequently, data addressing the secondary aim, to test how dog appetite and owner management are related, are discussed.

Questionnaire development

Factor structure

Analysis showed a clear factor structure which divided into 3 measures of dog eating behaviour, 4 of owner management and 2 related to health. Divisions were largely logical and self-explanatory from the items contained within.

Items regarding signs of gastrointestinal disease (3 items) formed a clear factor on analysis. Additionally, the ‘current disease’ category included two items which neither clustered with each other nor other factors. For this development work, further information was not collected about the nature of medical problems but this would be easily achieved using online or paper questionnaire formats if of relevance. Health questions were included because disease can influence a dog’s propensity to gain or lose weight either directly or secondary to drug therapy. Additionally, we considered that a dog’s display of food-related behaviour might be the net result of their inherent neurological drive to eat and physical factors such as nausea which could reduce food intake.

A particular concern was that subclinical or poorly recognised gastrointestinal disease (e.g., food allergy or chronic pancreatitis) might stop dogs with high food-motivation from expressing that behaviour. Presence of gastrointestinal signs did not affect body condition score or food-motivation, suggesting that should not be a great concern in future studies. The fact that regular vet visits or restricting exercise on vet advice was associated with higher body condition scores but no difference in food-motivation might reflect the effect of restricted exercise, medications or that obesity predisposes to disease (German, 2006; Raffan, 2013; Zoran, 2010).

We considered whether the lack of association between Current Disease or Gastrointestinal Disease factor scores and dog food-motivation scores might be due to the study being underpowered (since a fairly small proportion of dogs scored highly for either factor). However, a post hoc power calculation showed our study would have been powered to find a low correlation (R > 0.175) and the fact that neither factor was in actuality correlated significantly or with R > 0.04 means we feel the conclusions drawn above are valid.

Questionnaire validation

Several measures of the validity of the DORA questionnaire are reported and were found to be good. Analysis of reliability was assessed using answers by the same respondents made 3 weeks apart (test–retest reliability), a time frame which is on the cusp of being intra-rater reliability (which is usually assessed over a slightly shorter time frame) (Belshaw et al., 2015). Inter-rater validity was not measured in this study.

As a test of criterion validity, we hypothesised that dog factors would correlate with each other and dog food-motivation score and that similar would be true for owner management factors. The former was true. However, although each owner factor correlated with the overall owner management score (suggesting broad agreement amongst those constructs), it was notable that correlation between individual factors was less good. We concluded that it whilst it was valid to explore the data using the overall owner management score as we have at times in this paper, any future work designed to test how particular facets of owner management might affect dog obesity would be better served by considering each owner and management factor separately.

It is notable that, in the absence of a reference standard measure for dog food motivation, no measure of criterion (concurrent) validity could be applied to this questionnaire. Similarly, we chose not to collect in depth quantitative data about owner factors (e.g., number of walks per day, grams of food offered, nature of titbits). Responses to those types of questions have previously been shown to be unreliable and to fail to capture the complexity of different types of exercise or foods (German et al., 2011; Sallander et al., 2001; Slater et al., 1992). As a result, criterion (concurrent) validity could not be applied to management factors either. Instead, for both, we used semi-structured interviews with a subset of Sample 2 to check that what owners were answering in the questionnaire corresponded with more in depth verbal descriptors of each factor. The internal consistency of the items in these factors suggests that the approach adopted of capturing qualitative data has been successful.

Further validating this qualitative approach, the results confirmed some previously reported features of canine obesity that act as an external validation for the questionnaire: compared to lean dogs, obese dogs were more likely to be fed human food and titbits, have owners who were less likely to weigh/measure their food allowance, and exercise less (Bland et al., 2009; Courcier et al., 2010; Kienzle, Bergler & Mandernach, 1998; Robertson, 2003; Sallander et al., 2010; Warren et al., 2011). All those points have been discussed when first reported, and are intuitive. It is remarkable, however, that we have demonstrated those associations using just 16 relevant items, rather than using in depth, quantitative questions about each aspect.

Comparison with human eating behaviours

In comparison with similar human questionnaires, there was moderate similarity but fewer factors were identified (Llewellyn et al., 2011; Wardle et al., 2001). Several dog factors are directly analogous to those identified in studies of babies and children (e.g., ‘food responsiveness and satiety’ is analogous to the ‘food responsiveness’). However, dog owners only rarely recognised emotional under-eating or over-eating in their dogs, reported they did not feel fit to judge their dogs’ enjoyment of food, and confirmed that dogs universally had free access to water at all times, meaning other factors identified in children and babies are not relevant.

There was a similarly mixed picture for owner management of their dogs’ food intake. ‘Owner intervention to control weight’ mapped to ‘control over eating’, though many dog owners exert far more rigorous control than even strict parents. Giving human food to dogs is similar to instrumental or emotional feeding. Prompting/encouragement to eat is the final construct on the Parental Feeding Style questionnaire (Carnell & Wardle, 2007). During interviews, it was apparent that many dog owners added flavoursome items to their dogs’ regular food, but they appeared to have very variable motivation (e.g., ‘it uses leftovers’, ‘he won’t eat biscuits without gravy’, ‘it bulks him out without many calories’) and occur in dogs with different food-motivation. This proved too complex to capture on the DORA questionnaire and would be unlikely to inform the causes of canine obesity or define appetite so was not included.

In summary, the DORA questionnaire has proven to be a robust, easily applicable owner reported questionnaire to assess dog food-motivation and owner management. Following the validation, data was analysed to test the association between dog eating behaviour and owner management.

Variability in canine food-motivation and owner management

This is the first time food-motivation has been shown to vary considerably between individual dogs and between dog breeds. Dogs which were more highly food-motivated were shown to be more commonly overweight. Indeed, food-motivation had similar magnitude of effect to exercise and owners feeding their dogs human food in the minimum model for factor association with body condition score. To date, the literature, veterinary surgeons and surrounding debate has emphasised owner failure to control diet and exercise as the major reasons for dogs becoming obese (Bland et al., 2010; Degeling, Rock & Toews, 2011). However, this finding would suggest dogs’ individual drive to eat is as important as those factors.

The significant difference in food-motivation between breed groups is notable and, viewed alongside previously reported breed predispositions to obesity, suggests a genetic influence on food-motivation and obesity (Colliard et al., 2006; Edney & Smith, 1986; O’Neill et al., 2014). An alternative explanation would be that owners of particular breeds of dog somehow manage their dogs in such a way as to encourage food-seeking behaviour. However, the authors find it implausible that owners of particular breed groups would (a) have a tendency to feed and exercise their dogs in a similar way, and (b) that the differences between owners of different breeds would be sufficiently consistent to explain the variability in appetite seen here.

Furthermore, the finding from this study that owners of more highly food-motivated dogs exert greater control over their dog’s food intake directly contradicts the commonly held view that dogs only display food-seeking or begging behaviour because they have become habituated to receiving food from indulgent owners. Rather, it suggests owners generally make efforts to prevent the development of obesity as has been observed previously in parent/child relationships (Carnell et al., 2011; Fildes et al., 2015).

In contrast, there was no association between high food-motivation and exercise, the other way in which owners might limit their food-motivated dog’s tendency towards gaining weight. This likely reflects that the amount of exercise a dog receives is the net outcome of many conflicting factors: highly food motivated dogs are more likely to be overweight, which mean they would benefit from more exercise but also means they are likely to play less and exercise less vigorously (German et al., 2012), even if time-constrained owners tried to increase their exercise.

Limitations

This is an owner-report measure and could be subject to bias. The best available comparators for validation were the answers to questions during an in depth semi-structured interviews with respondents. Generally, there was concordance between questionnaire responses and interview results. In the few cases where there was discordance, those items were eliminated. Ultimately, the best way to validate the items relating to dog food-related behaviour would be comparison to food intake trials in large numbers of pet dogs, but this is not feasible at present though that approach has been used to validate the Child Eating Behaviour Questionnaire and will be considered in future (Carnell & Wardle, 2007). Similarly, owner management and exercise items might best be compared to longitudinal monitoring using food diaries, video monitoring or accelerometry (Chan et al., 2005; Slater et al., 1992).

The measure of obesity used was an owner-assessed body condition score. Dogs of different breeds are very heterogeneous so weight is a poor indicator of obesity in the species. In contrast condition scoring according to body morphometry following visual inspection and manual palpation is well validated (Laflamme, 1997). However, it is well recognised that owners tend to assign inaccurate body condition scores to their dogs, usually assigning a score closer to the ‘ideal’ (Colliard et al., 2006; Courcier et al., 2011; White et al., 2011).

Consistent with this, 23% of owners in this study assigned their dogs body condition score 4 or 5/5 (overweight/obese) despite the fact that estimates suggest obesity prevalence is at least 25% in the general dog population and many estimates suggest as high as 40% (German, 2006; McGreevy et al., 2005; O’Neill et al., 2014; Zoran, 2010). Thus our data may not be robust enough to compare the actual body condition scores to other studies, but the number of respondents and normal distribution of body condition scores across the sample means it is reasonable to use the scores to interrogate the data in the limited way demonstrated here. Furthermore, it is notable that clear associations between different factors (dog food-motivation, restriction of human food and exercise) and body condition score were identified, despite the fact that owner’s tendency to underestimate the body condition score of overweight dogs would have been predicted to reduce the power of the study to detect such associations. Future work using more reliable, vet/nurse assigned body condition scores would be of value.

Future applications

The authors envisage that this questionnaire might be of value to others both in clinical practice and research. Clinically, enumeration of food-motivation might help motivate owners to work hard to restrict food and reduce some of the ‘blame’ often assigned to owners of obese dogs; this would be an interesting future study to perform. Clinical researchers might use the questionnaire to discover if management practices altered food-motivation, though application to a longitudinal study would require further validation as this work did not examine if a dog’s score would change over months or years. Biomedical researchers using dogs to model obesity in other species might find value in being able to quantify food-motivation as part of studies to test the effect of genes, drugs or other variables on food intake and obesity.

Conclusions

We have developed and validated a robust, practical, user-friendly, owner-reported measure of dog eating behaviour and of management and health factors with potential to affect the development of obesity in dogs. It will be a useful tool for research into obesity and related behaviour, and in veterinary clinical practice. We have demonstrated that there is marked variability in canine food-motivation both between individuals and breeds. Highly food-motivated dogs are more likely to be obese, with food-motivation having as great an association with obesity as owners giving titbits and exercise on body condition score. Finally, owners of highly food-motivated dogs tend to exert greater control over their dogs’ food intake.

Supplemental Information

Supplemental Information 1 Raw Data from DORA questionnaire

NA, not answered F, M, FN, MN, female, male, female neutered, male neutered (neuter status shown where given) Questions in order presented on questionnaire. See right hand columns for collated scores for factors and overall dog/owner. These are the data from sample 2, as described in the main body of results section in Raffan et al., DORA questionnaire description.

Click here for additional data file.

File S1 Semi-structured interview guide

Click here for additional data file.

File S2 Final DORA questionnaire

Click here for additional data file.

Additional Information and Declarations

Competing Interests

Author Contributions

Animal Ethics

The authors declare there are no competing interests.

Eleanor Raffan conceived and designed the experiments, performed the experiments, analyzed the data, contributed reagents/materials/analysis tools, wrote the paper, prepared figures and/or tables, reviewed drafts of the paper.

Stephen P. Smith analyzed the data, contributed reagents/materials/analysis tools, prepared figures and/or tables, reviewed drafts of the paper.

Stephen O’Rahilly conceived and designed the experiments, contributed reagents/materials/analysis tools, reviewed drafts of the paper.

Jane Wardle conceived and designed the experiments, reviewed drafts of the paper.

The following information was supplied relating to ethical approvals (i.e., approving body and any reference numbers):

The study was approved by the Ethics and Welfare Committee of the Department of Veterinary Medicine, University of Cambridge (CR125).

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
