# Peer review of "Development, factor structure and application of the Dog Obesity Risk and Appetite (DORA) questionnaire"

_PeerJ, doi:10.7717/peerj.1278_

## Round 0.1 · original submission · Major Revisions

Both reviewers see the value of the work conducted and I agree that this is an interesting and potentially useful contribution to the field. Both reviewers make suggestions designed to improve clarity. The reviewers vary slightly in their suggestions for restructuring but I would like to see you adopt the approach suggested by Reviewer 2 with a clearer distinction between Methods and Results. This can be done in tandem with Reviewer 1's suggestion of a flow diagram to explain the process adopted.

I agree with both reviewers that it is important to include the full version of the final questionnaire - as essentially this is the main "result" of your study.

Reviewer 1 also makes an important point relating to the second aim of the study - testing how reported differences in appetitive behaviour in dogs are associated with obesity or condition score, given the rather small number (and likely underestimated number) of dogs that were reported to be overweight. A similar issue arises with your conclusion that the presence of gastrointestinal signs did not affect condition score or food-motivation (Line 356). Some indication of the power of your data to detect such effects is needed.

Please clarify the use of the term "theme" (Reviewer 1). Also "emotional eating" - as this term is used in various ways in the human literature.

There are a number of places, described by both reviewers, where test statistics or other statistical information is missing or could be clarified.

Finally, I have a few minor points that I'd like you to consider:

Generally, in the main body of the paper you have been careful not to make unwarranted cause/effect extrapolations but there are a few places where this creeps in e.g. abstract line 19; line 39 - variation in appetitive behaviour may be associated with obesity and owner management but it does not necessarily influence it. The questionnaire asks for cross-sectional not longitudinal information so (for example, and however unlikely) we cannot know whether some owners buy fat dogs and then find that these dogs are reluctant to exercise. Food motivation of fat dogs may also increase as their energy needs will be greater for the same activity level. In other words the current study says little about obesity 'development'.

line 55 variation...is

line 74 - positive reinforcement is a well-defined term and does not need inverted commas - could write as ...food is commonly used as positive reinforcement during training

line 80-84 - you could use this to better define your own study - presumably all of these other studies have asked about obesity but not about eating behaviour and motivation? This could be made clearer early on.

line 109, 111, 113 - it is not quite clear who is classifying behaviours as desirable or undesirable, or advising use of food as training aid - add the specific references to the texts. How relevant were these classifications for your study?

line 178 - Facebook and Twitter sites?

line 181, 250 - do you know why this bias towards labradors - due to social media spread amongst this group? State estimated proportion of labradors in UK dog population so extent of bias can be assessed.

Line 198 - please define complexity and communality.

Line 256 - additional information about omitted questions would be useful - how many were eliminated for each of the 3 reasons given?

Any commonalities in the questions eliminated for high complexity, for example?

Discussion - be careful not to simply restate methods/ results (e.g. lines 332-340).

Conclusions - you have not reported any results on user-friendliness. I think that you don't know how many potential participants may have started to answer the questionnaire but then not completed it because they did not find it user-friendly.

Line 456 - again be careful about a conclusion that implies that food-motivation influences (causes) obesity - see comment above.

Table 1 and elsewhere - should Dog Factor 3 be called Interest in Titbits?

·

Basic reporting

I found the content of this manuscript difficult to follow in its current format. The authors have followed a necessarily complex, multi-step iterative process to design, validate and produce results from a novel questionnaire. I felt it was challenging to find the results which had informed the next step. A combined methods and results section for each of questionnaire design, validation and results generation steps may simplify things if the journal would allow that.

In addition, clarity would be aided by inclusion of the following:
a) A copy of the semi-structured interview question guide. The authors refer to "themes" but no reference is made to any analysis (e.g. grounded theory, thematic analysis) which was done to generate these from what must have been a considerable amount of interview data. Themes in qualitative research are typically derived from analysis and the use of the term in this manuscript is currently ambiguous. In addition, was a sampling frame used? Why 50 interviews, it seems like a huge number - was data saturation not reached until this point? How were the interviews recorded and transcribed? Was this done verbatim? Was any software used to aid analysis or was all the analysis done just on the notes made by the interviewer at the time? This is all important information which is currently not reported.
b) Inclusion of a complete reproduction of the final validated questionnaire clearly showing all the questions and response options. My personal opinion is that a publication about the validation of a new instrument is of little value to the scientific community if that instrument is not then available for use by others. I appreciate that this is an online questionnaire, but a screenshot of the layout should be a minimum included along, ideally with a PDF version of the questionnaire or a link to a useable online format. I can see the raw data are available as a supplementary file, but this will be of little use to those who wish to use the questionnaire for their own research. In addition, more information should be provided about how that questionnaire was presented to dog owners - was it called the DORA at that point, were owners aware that this was a questionnaire about obesity? This is important as it may be a potential source of bias.
c) A flow diagram clearly showing what was done at each step of the validation process would be incredibly helpful. I became lost when the authors introduced scores and composite scores. Since the results are reported using these terms, it is vital that their derivation is clear.

Experimental design

I am not an expert in the methods used to validate this questionnaire and strongly recommend that someone with expertise in questionnaire validation be asked to review the materials and methods and results section. Simple inclusions such as stating a significance value for p, and specification of the test used for normality would be useful.

Validity of the findings

I cannot easily comment on the validity of the findings due to my lack of expertise in the methodology of instrument validation. I would be interested to have the opinion of a statistical reviewer as to whether the strength of the conclusions is valid given that only 10% of dogs included in the validation of this questionnaire were obese as reported by their owners, and given the acknowledged limitation that it is possible that up to 15% of other dogs included in this validation may also be obese but their owners not have classified them as such. The authors conclude that this tool will be of use to clinical veterinarians. I am uncertain how clinicians will find it useful - could the authors substantiate this statement? How will it be useful in the future in obesity research, in particular in relation to the dog as a model for human obesity?

Additional comments

This is interesting work - obesity is a big problem in the canine population and identification of factors which lead to it are helpful. It would be useful if the authors could put their results into context - what is the practical relevance of this research to veterinarians and dog owners dealing with an obese pet? If DORA is not to be made available to the public, the significance of results of the questionnaire are arguably more relevant than its factor structure. I am sure the authors are aware that questionnaire validation is a complex multi-step process. As far as I can see, the authors have performed test-retest reliability, internal consistency and content validity on this questionnaire. Additional tests of validation which would be useful to perform in the future would be inter and intra-rater reliability and criterion validity.

Reviewer 2 ·

Basic reporting

I think this is a very worthy and interesting manuscript which should be published following revisions. The study is well conducted and much thought and consideration has gone into the design of this questionnaire. I think all PeerJ polices have been upheld.

Experimental design

The study is well conducted and through in its design.
The main revisions I suggest relate to the presentation of the methods and results and some small additional statistical analysis would also be recommended.
With regard to the methods, these are currently presented in a chronological way detailing what was achieved. I would like to see this restructured and/or rewritten according to 1) the questionnaire, sample, method of dissemination; 2) statistical methods. The latter should be divided according to the type of development or validation that was assessed. The authors need to be clearer on what they have and haven’t tested and could partition the methods according to the test they have conducted. It would be good if subheadings about tests conducted could match between the methods and results. They also need to state what accepted values for each test would be and what they consider good, low etc. for each test. You should state at the end of the introduction which tests of validation you will apply. Table 2 in this article may provide a useful checklist for checking which tests of validation have been conducted in this study:
http://ac.els-cdn.com/S109002331500307X/1-s2.0-S109002331500307X-main.pdf?_tid=dcb5adea-375c-11e5-b705-00000aab0f6b&acdnat=1438330948_472d003054f94bebcbdc2f3b8513498d
I would like the test of body condition reframed as criterion validity. I don’t think construct validity has been performed and this would be relatively simple to test. Construct validity is where hypotheses are made about which questions (or groups of questions) will be correlated and the direction. Then these are tested. This is distinct from internal consistency which I think has been tested. With regard to internal consistency, I would like to see ICCs or Kappa coefficients or similar presented for each grouping of questions. I can’t find a description of how test retest reliability was analysed. I would like some more details on the choices made about the PCA and justification of these choices. E.g. You need to outline apriori what you would do if you found a question that loaded on more than one factor. You need to say why you chose the covariance structure you did. etc This paper is a good reference for this:
http://onlinelibrary.wiley.com/doi/10.1111/j.1439-0310.2010.01758.x/abstract
I think you didn’t test intra or inter observer reliability which is OK but need stating somewhere. I am not convinced about presenting data on different breeds etc, in terms of fit with the paper. How does this relate to the validity of the questionnaire? Perhaps this is data to present elsewhere after the presentation of the validation of the questionnaire? If this is to show the questionnaire can be widely applied make this more obvious from the start.
There is some confusion between methods and results with some results (like number of questionnaires received) in methods rather than results. The response rate achieved is a finding. Similarly there are cases where there are methods in the results e.g. line 271 three weeks later owners complete the questionnaire. This should be reframed as a finding. E.g. the response rate for test retest was X with 79 owners completing the questionnaire 3 weeks after completing their first completion.

Validity of the findings

I think the study conforms to PeerJ policies in this regard.

Additional comments

An overall point is that I would strongly encourage consideration of the final developed questionnaire to be presented in full. This is of course an author and editorial decision but I think it helps other researchers to continue or develop the research if the questionnaire is available in an easy to read format rather than having to decipher the questions that should be asked from various tables in the paper. This will also help citations. Given the journal is online format there shouldn’t be restrictions on this due to the length of the paper.

Minor comments:
Line 30 write four not 4
Line 35 are you missing a word here? Should this be: More highly food motivated dogs exerted…?
Lines 43-44. First line should have some references to support the points.
Line 57 Suggest rephrasing
Line 66 behaviour doesn’t need as s (behaviour is what an animal is doing so doesn’t need an s there are separate types of behaviour)
Line 70 Rephrase. I don’t think this makes the point you want it to as it is presently worded. Do you mean that questionnaires that have been developed to study baby and children’s eating patterns have utilised theories on eating styles. Since eating style theories do not exist for dogs these were inferred from animal care textbooks etc? Could you expand on what eating styles these texts infer in dogs?
Line 85. Be clear about what kinds of validation you have achieved. Full validation for me would be predictive criterion validity where you could predict whether a dog would get fat based on this questionnaire (accounting for modulating owner/management factors).
Line 99. Questionnaires designed to study eating behaviour in children and babies not adult humans- make the distinction.
Line 106- whose clinical experience? The authors, other vets?
Line 107 behaviour was variable (not were)
Line 118. I think some more detail on the questions asked would be useful
Line 123-132. This is slightly oddly worded for qualitative work. This is not my area of expertise but I have seen terms like non-judgementally and pejorative used to refer to interview methods.
Line 160-161. This doesn’t sit well here. You could perhaps comment on this in the discussion.
Line 162. Could provide a reference here for how you analysed and kept notes.
Line 217 this is not the correct reference for R if you type citations() into R you will get the correct reference
Line 222 missing a space after bracket
Line 573 you don’t state the test used here unlike other tables.
Line 605. I wonder based on this graph if BCS of 5 should be included in analysis as there are few points here.

---

## Round 0.2 · accepted · Accept

Thanks for your thorough review. I have read your response carefully, and can see that you have taken account of the significant points raised by the reviewers. The manuscript is greatly improved and I am now happy to accept it.